# Data clustering to optimise the representativity of observational data in air quality data assimilation: a case study with EURAD-IM (version 5.9.1 DA)

Alexander Hermanns[1,2], Anne Caroline Lange[1], Julia Kowalski[3], Hendrik Fuchs[1,2], and Philipp Franke[1]

[1]Forschungszentrum Jülich, Institute for Energy and Climate Research: Troposphere (ICE-3), Jülich, 52425, Germany
[2]Faculty of Mathematics and Natural Sciences, University of Cologne, 50923 Cologne, Germany
[3]Faculty of Mechanical Engineering, RWTH Aachen University, 52062 Aachen, Germany

**Correspondence:** Philipp Franke (p.franke@fz-juelich.de)

**Abstract.** In the field of air quality analysis, data assimilation is commonly used to integrate information on the atmospheric state provided by observations into the model. However, the analysis is largely dependent on the data available to the assimilation system. In order to obtain an accurate analysis of the true state of the atmosphere, the representativity of the utilized data becomes a fundamental requirement. Here, a method is presented that derives a representative split of the ground-based monitoring network data that depends only on the characteristics of the observation data. The core of the methods is a clustering algorithm to subdivide the data into subsets. Two clustering algorithms, k-means, and k-mean soft constraint, are tested and applied to air pollutant observations in Europe. The clusters are solely derived from observation intrinsic properties (such as geographic location and averaged concentrations). The resulting clusters reliably distinguish common features of the observational data, e.g. mean and variance of averaged air pollutant concentrations. Representativity of the observational data in the assimilation and validation subset is ensured by sampling each cluster individually. The method is tested using the assimilation system of the chemistry transport model EURAD-IM (EURopean Air pollution Dispersion – Inverse Model) and evaluated for data from four months in 2016. A significant improvement of the analysis' representativity, quantified by the difference between the analysis' root mean square error with respect to the assimilation and validation dataset, is found in the results. Compared to an operational configuration, the largest improvement in the relative representativity measure is evaluated for CO with $16\,\%$, for $NO_2$ with $4\,\%$, and for $O_3$ with $1\,\%$. A reduction in the relative representativity measure is observed for $SO_2$ with $-5\,\%$, for $PM_{10}$ with $-2\,\%$ and for $PM_{2.5}$ with $-5\,\%$, although these differences do not lead to significant deviations in absolute values given the overall error and the improvement for CO outweighing the changes in the other species.

## 1 Introduction

Air pollution is the main environmental health risk (World Health Organization, 2021). Therefore, its assessment and reliable prediction are essential to further develop policies aiming at improved air quality. Although observations of air pollutants via air quality monitoring networks, mainly consisting of ground based in situ observations, are routinely used to monitor and evaluate air pollution, they only show a subset of the atmospheric state. A common approach is to comple-

ment the observations by model simulations, which show a more complete representation of the atmospheric state but lack of accuracy due to insufficient input data (e.g., initial values, boundary values, and emission data). Data assimilation is a method that integrates information from observational data into the model state with the objective of obtaining an analysis that is, in theory, the best available representation of the atmospheric state. However, this requires representative observations to be assimilated in order to obtain more accurate analyses with respect to the (unknown) true atmospheric state and atmospheric parameters, such as air pollutant emissions. The Copernicus Atmosphere Monitoring Service (CAMS, https://atmosphere.copernicus.eu/regional-air-quality-production-systems (last access: 07 August 2025)) employs an ensemble of air quality models using data assimilation to provide daily air quality forecasts and reanalysis over Europe. An overview of the ensemble and the models used within can be found in Marécal et al. (2015) and Colette et al. (2025). In data assimilation applications, observational data are typically split into an assimilation and a validation set, where the former is used for the data assimilation procedure and the latter is withheld and used to evaluate the analysis with independent observations. This split of the available data is known as sub-sampling. Each subset of the observational data must be representative of the atmospheric state to obtain the highest accuracy of the analysis.

Various measures to estimate the representativity of observational data have been explored, yet no consensus on a universal approach has been reached (Kracht et al., 2017). Nappo et al. (1982) developed a general definition for representativity for meteorological applications: "a point measurement is representative of the average in a larger area (or volume) if the probability that the squared difference between point and area (volume) measurement is smaller than a certain threshold more than $90\%$ of the time". This definition is employed in the work of Piersanti et al. (2015) to derive areas of representativity for air quality monitoring stations in Italy and to show their seasonal variability. Other methods determined the redundancy of air-quality observations in urban areas and their correlation with satellite data to define the representativity. For example, Ma et al. (2019) found $PM_{10}$ (particulate matter with an aerodynamic diameter of $10\,\mu m$ or less) to be the primary air pollutant in Lanzhou, China, from July to December 2015, and identified the most representative observation site from a group of four. Su et al. (2022) analysed the spatial representativity of air quality monitoring networks estimated using environmental influence factors on $PM_{2.5}$ concentrations (particulate matter with an aerodynamic diameter of $2.5\,\mu m$ or less). With this, they applied k-means clustering to identify optimal locations for additional monitoring stations in under-represented areas, which resulted in additionally 40 monitoring stations and a $15\%$ increase in the representativity. Ignaccolo et al. (2008) converted discrete time series of pollutant observations in the Piermonte region, Italy, into spline coefficients and identify similarities using a Partitioning Around Medoid (PAM) clustering algorithm. They were able to show different spatial patterns in the clusters for various species depending on anthropogenic and geographical influences. The study by Joly and Peuch (2012) characterizes pollutant time series by 8 parameters derived from their daily-, monthly-, seasonally- and annually averaged diurnal cycles. With this, they were able to derive 10 classes of air pollution monitoring sites using linear discriminant analysis, which allow for the classification of new sites without the need to recalculate the classification. This classification is used by the Copernicus Atmosphere Monitoring Service (CAMS) to filter observational data in order to improve their regional products (Peuch et al., 2022).

In contrast to most literature, this study focuses on generating a representative sub-sampling of the available observational data instead of assessing the representativity of individual observational data directly. For this, a method for the generation of the sub-samples is developed that bases on data clustering and grouping the available observation stations according to their characteristics. The advantage of this data based approach is that it can be applied to the input data of the assimilation system and does not require prior model evaluations. Here, the relative representativity of two datasets is determined by quantifying and subsequently comparing the difference in the RMSEs between the model analysis and the observations from the assimilation and validation data set. This measure is hereafter called AV-difference and is described in detail in Sec. 3.4.

Cluster analysis is a statistical data analysis method aiming to identify subgroups (clusters) of similar data points (objects) within a data set based on their features. The similarity of objects can be measured as the mutual distance in feature space, where the exact distance measure may vary depending on the employed method. Objects in the same cluster are considered more similar to each other than to objects in other clusters. Cluster analysis has been applied in atmospheric applications for various purposes. Some examples include the improved identification of patterns of air quality regime classifications for ozone ($O_3$), nitrogen dioxide ($NO_2$), sulfur dioxide ($SO_2$), and $PM_{10}$ for observation stations in Germany (Flemming et al., 2005). Lyapina et al. (2016) clustered European ozone observations based on monthly averaged diurnal cycles and showed that different $O_3$ regimes (i.e., groups of observation stations with similar statistical characteristics) are not geographically separated. Carro-Calvo et al. (2017) identified zones with coherent spatio-temporal patterns in observations of summer ozone concentrations and were able to link weather patterns to the occurrence of extreme values. Utilizing temperature and precipitation, Fovell and Fovell (1993) were able to identify four climate zones in the United States and further divided them into subzones by using a higher number of clusters. A review about clustering applications and methodologies in air pollution studies from 1980 to 2019 can be found in Govender and Sivakumar (2020).

As a common clustering algorithm, k-means was applied to derive correlations between $O_3$ and $NO_2$, nitrate radical ($NO_3$), night-time nitrous acid (HONO), and formaldehyde (HCHO) by (Wang et al., 2022). In addition, (Borge et al., 2022) used k-means to identify zones of common air quality in Madrid. On global scales, the method was used to analyse aerosol regimes (Li et al., 2022).

## 2 Chemical transport model EURAD-IM

The EURopean Air pollution Dispersion – Inverse Model (EURAD-IM Elbern et al., 2007; Franke et al., 2024) is used to evaluate the effect of different sampling strategies of observational data in a realistic modeling framework. EURAD-IM is a chemistry transport model capable of three- and four-dimensional variational (3D-/4D-var) data assimilation applied for air quality evaluation (De Souza Fernandes Duarte et al., 2021; Tillmann et al., 2022; Liu et al., 2024) and analyses (Franke et al., 2022; Erraji et al., 2024). Here, 4D-var is used for jointly optimizing initial values and emission factors (Elbern et al., 2007). The utilized adjoint model comprises all relevant processes that are included in the forward model, namely advection, horizontal, and vertical diffusion, chemical transformation, and secondary inorganic aerosol formation. The model domain covers Europe with a $15\,\mathrm{km} \times 15\,\mathrm{km}$ horizontal resolution and includes 30 vertical layers up to $100\,\mathrm{hPa}$ with the additional option of nesting

domains (5 km × 5 km for central Europe, 1 km × 1 km for smaller domains). Meteorological input data are retrieved from the Weather Research and Forecasting Model (WRF) version 3.7 (Skamarock et al., 2008) with Integrated Forecasting System (IFS) forecast data (Inness et al., 2019) taken as initial and boundary values. Chemical boundary values are provided by the complementary chemistry modules of the Composition Integrated Forecasting System (C-IFS). Anthropogenic emissions are taken from the German Environment Agency (Submission 2019, version 2) for Germany, due to the available data being directly

aggregated onto the model grid, and from the Copernicus Atmosphere Monitoring Service's regional anthropogenic emission inventory (CAMS-REG-AP, version 3.1) for Europe. It is worth noting that the proposed sampling method is independent of the used data assimilation model and the EURAD-IM simulations solely serve as a validation of the proposed method.

## 3    Cluster analysis

The observational data used in this study are ground-based observations used within CAMS regional analysis. The data is pre-

filter to the classes 1-7 according to Joly and Peuch (2012) from data accessible at the European Environmental Agency (EEA) via

https://eeadmz1-downloads-webapp.azurewebsites.net/ (last access: 09 June 2025). They consist of measurements of hourly concentrations of carbon monoxide (CO; 382 stations), nitrogen dioxide ($NO_2$; 1743 stations), ozone ($O_3$; 1712 stations), sulfur dioxide ($SO_2$; 1005 stations), $PM_{10}$ (1029 stations), and $PM_{2.5}$ (509 stations) from 40 EEA member and cooperating

countries. In total, 6380 individual time series from 2279 stations are used after filtering the data as outlined in Sect. 3.3. An overview of the geographic distribution of the available observation stations for each species is shown in Fig. 1. The density of CO observations compared to the other measured species is substantially lower, even in highly populated areas.

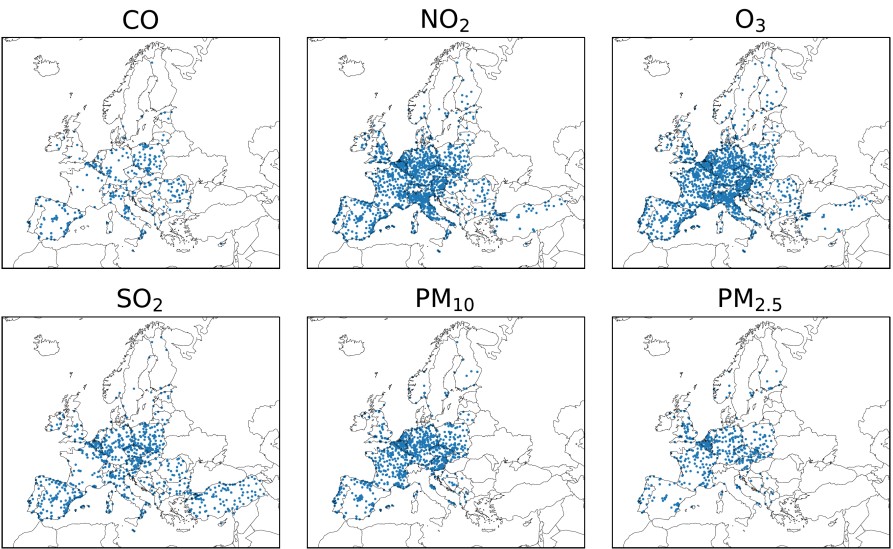

**Figure 1.** Distribution of available observation stations per species in Europe in 2016.

The clustering process divides observation stations into subgroups (clusters), determined by the algorithm itself and the features used. Finally, by sub-sampling the clusters into assimilation and validation data, the different observational features can be included equally in both data sets. The representativity of these sets obtained by the clustering is analysed in this work. An overview of the methodology is shown in Fig. 2, where the individual components are detailed below.

## 3.1 K-means algorithm

In this study, clustering algorithms based on k-means (MacQueen, 1967), sometimes referred to as the Lloyd–Forgy algorithm (Forgy, 1965), are employed. K-means aims to split a number of $n$ objects $(d_1...d_n)$ with feature vectors $\boldsymbol{f}(d_i) \in F$, elements of the vector space $F$, into $K$ clusters following the algorithm outlined below:

1: Initialize K random cluster centres (centroids) $C_1...C_K$ as elements in $F$.

2: For all $i$, assign each object $d_i$ to the closest cluster $C_J$ by minimizing the squared Euclidean distance, Eq. (1).

3: For all clusters, compute the new cluster centres $C'_J$ by averaging the feature vectors within the cluster.

4: Iterate step 2 and 3 until the cluster centres $C_1...C_K$ do not change or until a fixed number of iterations is completed.

In the application of this work, objects are observation stations and the features are the station characteristics (e.g., latitude, longitude, altitude, and observed concentrations of the different species). The squared Euclidean distance between the features of $d_i$ and the centroid $C_J$ is defined as

$$dist_{euclid}(d_i, C_J) = \|\boldsymbol{f}(d_i) - \boldsymbol{f}(C_J)\|^2, \tag{1}$$

where $\|\cdot\|$ is the Euclidean norm. The elements of the feature vectors are normalized to a dimensionless value between $[0,1]$ in order to ensure that each feature affects the distance value with the same weight. The number of clusters $K$ is chosen before the computation. Different approaches exist to find the optimal number of clusters. In this study, the cluster hyperparameters (i.e. number of clusters $K$) are determined by selecting those that maximize the mean silhouette score $\bar{s}$, as defined by (Rousseeuw, 1987),

$$\bar{s} = \frac{1}{\#d_i} \sum_{d_i} \frac{b(d_i) - a(d_i)}{max\{a(d_i), b(d_i)\}}, \tag{2}$$

where $a(d_i)$ is the averaged distance of the object $d_i$ to all other objects in the same cluster $C_I$ and $b(d_i)$ the averaged distance of $d_i$ to all elements in the next closest cluster, $\#d_i$ is the total number of objects $d_i$.

For $d_i \in C_I$,

$$a(d_i) = \frac{1}{\#C_I - 1} \sum_{d_j \in C_I} dist(d_i, d_j), \tag{3}$$

$$b(d_i) = \min_{J \neq I} \frac{1}{\#C_J} \sum_{d_j \in C_J} dist(d_i, d_j), \tag{4}$$

with $\#C_I$ denoting the number of objects in the cluster $C_I$. Although different choices to calculate the distance are possible, it is convenient to select the Euclidean distance Eq. 1 as distance functions in Eq. 3 and 4. The mean silhouette score is an

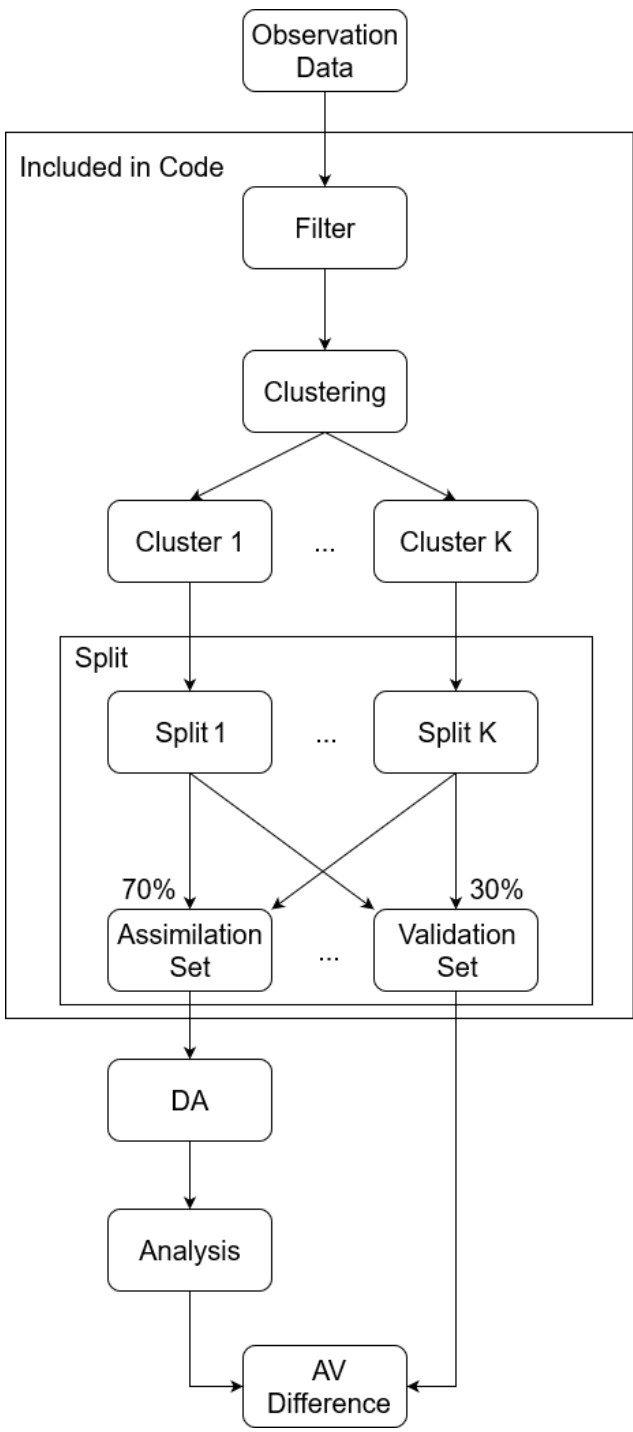

**Figure 2.** Flowchart of the proposed methodology. In the presented case, 'Observation Data' is the observation data used in the CAMS-project. The elements in the box titled 'included in code' are included in the provided software code (Hermanns, 2025), with the KSC algorithm as the 'Clustering'. The box 'split' is the extraction of the assimilation and validation set from a clustering result. 'DA' is short for data assimilation. The 'AV-difference' is the representativity measure and detailed in Sec. 3.4.

effective measure for the quality of the clustering. It returns values from $[-1, 1]$ with $\bar{s} \approx 1$ indicating an 'optimal' clustering. For $\bar{s} \approx 0$ the result is inconclusive, since objects are equally distant to their assigned and second closest cluster, and $\bar{s} \approx -1$ indicates that $d_i$ is assigned to the wrong cluster.

## 3.2 K-means soft constraint algorithm

The concentrations measured at an observational site contain valuable information in order to determine the similarity to other stations. Since not all species are measured at all stations, certain feature vectors do not contain valid values for all entries. Therefore, the k-means soft constraint algorithm (Wagstaff, 2004) is used for the clustering. It aims to identify clusters in data, where not all features have values for all objects. This avoids the need to fill-in missing values by interpolated values, which can lead to inaccuracies due to necessary assumptions. In the following, the method by Wagstaff (2004) is described.

The vector space of feature vectors $F$ is separated into the features that are known for all objects, $F_o$, and the features that are missing in some objects, $F_m$, Such that $F = F_o \oplus F_m$. Clustering is then performed for $F_o$ with constraints based on $F_m$. The constraint $\sigma(d_i, d_j)$ indicates the separation between the objects $d_i$ and $d_j$ in the subspace $F_m$. It is defined using the features $\boldsymbol{f_m}(d_i)$ and $\boldsymbol{f_m}(d_j)$ from the vector subspace $F_m$, with

$$\sigma(d_i, d_j) = \sqrt{\sum_{\boldsymbol{f_m} \in F_m} (\boldsymbol{f_m}(d_i) - \boldsymbol{f_m}(d_j))^2}. \tag{5}$$

Features that are missing in both objects do not contribute to the constraint. The k-means soft constraint algorithm consists of the same steps performed in the classical k-means algorithm. However, it uses a modified distance function

$$dist_{KSC}(d_i, C_J, \omega) = (1 - \omega) \frac{dist_{euclid}(d_i, C_J)|_{F_o}}{V_{max}} + \omega \frac{\Gamma_{d_i}}{\Gamma_{max}}, \tag{6}$$

with $V_{max}$ being the maximum variance considering all objects. $\Gamma_{d_i}$ is the sum of all squared constraints for $d_i \in C_I$ with objects in the same cluster $d_j \in C_I$,

$$\Gamma_{d_i} = \sum_{d_j \in C_I} \sigma^2(d_i, d_j). \tag{7}$$

$\Gamma_{max}$ is the sum of all squared constraints whether $d_j$ is in the same cluster as $d_i$ or not, i.e., $\Gamma_{max} = \sum_{d_j} \sigma^2(d_i, d_j)$. Furthermore, a scaling parameter $\omega$ is introduced, weighting the influence of $F_m$ in relation to $F_o$. The first term of Eq. (6) is the squared Euclidean distance normalized by the maximum variance. Thus, for $\omega = 0$, the k-means soft constraint algorithm leads to the same result as a standard k-means algorithm in which only features in $F_o$ are included. The second term increases the distance of $d_i$ to its assigned centroid $C_I$ scaled by the constraints. It is important to note that this limitation prevents the algorithm from converging in many applications due to an artificially increased distance between objects within the same cluster compared to objects in other clusters.

## 3.3 Data preparation and experimental design

Before the application of the data clustering to observational data, the data has to be prepared to avoid influence of outliers which can negatively effect the clustering quality (Nowak-Brzezińska and Gaibei, 2022). This is achieved by filtering the data with empirically determined thresholds. Values below $0\,\mu g\,m^{-3}$ or exceeding $50000\,\mu g\,m^{-3}$ for CO and $5000\,\mu g\,m^{-3}$ for all other species are removed from the observation data to exclude outliers. Time series for species with a relative standard deviation below 0.01 in the assimilation period are excluded, as it is assumed that the hourly concentrations of air pollutants usually have a greater temporal variability. Moreover, observations are removed for intervals where consecutive observations are constant for more than two hours. A summary of the filtering procedures and their effect on the amount of observational data is shown in Table 1.

Two different classes of features have been used to cluster the observational data. The first set of features in the clustering algorithms are the mean and the variance of the annually averaged diurnal cycles for each species. While the diurnal cycles vary throughout the year and can exhibit different behaviours depending on the season, weather conditions or human behaviour, an average over the entire year is not susceptible to short time-scale variations. This is done to better compare the test configurations with the reference, which is the configuration used in the CAMS reanalysis for 2016 (discussed later on). The placement of observations in the reference is static for the entire year and to ensure a fair comparison, the test configurations and thus their clustering features are also chosen to be static. The mean and variance of the annually averaged diurnal cycle are chosen to characterize the differences in the measurement sites. While other features are also suitable for clustering as was shown e.g., in Joly and Peuch (2012), restricting the number of features is important to apply meaningful clustering. In high dimensions, the "nearest neighbour" problem cannot be solved in all cases (Beyer et al., 1999).

The annually averaged diurnal cycle is calculated from the filtered observational data by computing the average for each hour of the day for the entire year. From this annually averaged diurnal cycle, the mean and the variance are calculated and form one set of the features used in the clustering algorithms. A filter is implemented where the standard deviation of these values is calculated for each species individually and measurement stations where the mean or the variance of the annually averaged diurnal cycle exceed 7.5 times the species-specific standard deviation are excluded.

The second set of features used in the clustering algorithms are the geographic coordinates, which are first converted to the Cartesian coordinate grid used in the EURAD-IM based on a Lambert conformal grid projection. To ensure that all features have the same weight, each feature is separately normalized to an interval of $[0, 1]$.

Two experiments are performed to evaluate the two clustering algorithms on the data. The first experiment applies the k-means clustering algorithm and is named Clustering-Diurnal (CD) in the following. The features consist of the mean and variance of the annually averaged diurnal cycle of CO, $NO_2$, $O_3$, $SO_2$, $PM_{10}$, and $PM_{2.5}$ observations from 2279 air quality monitoring stations while the measurement stations themselves are the objects. The feature vectors contain a maximum of 12 valid elements for each object. Since not all stations provide measurement data for all species, missing values are ignored for the calculation of the cluster centres. In the second experiment, the k-means soft constrained algorithm is applied, named KSC in the following. Here, the features are the geographical coordinates and the altitude for each measurement station as well as

the mean and variance of the annually averaged diurnal cycle. The geographical coordinates and altitude are available for all objects (i.e., the measurement stations) and thus correspond to the features from the feature space $F_o$ (cf. Eq. 6). The mean

and variance of the annually averaged diurnal cycle are not available for all species for all objects and are used as constraining features of the feature space $F_m$. As the validation of a species at a measurement station may be influenced by the assimilation of measured concentrations of other species at the same location due to their chemical coupling, all observations from one measurement station are assigned to either the assimilation or the validation set (data bundling).

The optimal parameters used in both clustering algorithms are determined by maximising the silhouette score (Eq. 2).

The number of clusters $K$ is varied from 2 to 9 and the k-means soft constrained parameter $\omega$ from 1 to 0, with decreasing increments towards higher silhouette scores. In the case of the KSC experiment, the algorithm is not deterministic and $\bar{s}$ can vary depending on the initial choice for the cluster centers. For the CD experiment, the optimal number of clusters is $K = 2$ and for the KSC experiment, $K = 8$ and $\omega = 1.0 \times 10^{-6}$ are found to generate the largest silhouette score.

To ensure the equal representation of each cluster in the validation and assimilation data sets, the data split is applied ran-

210 domly on each individual cluster. With this, $70\,\%$ of the data is assigned to the assimilation data set, while the remaining $30\,\%$ are taken as validation data. The data split is designed to ensure a random 70/30-split for each species in each cluster while maintaining the data bundling. The resulting data splits are referred to as observation configurations in the following sections. To verify that no unintended correlations between the assimilation and validation data sets have been introduced, which could affect the clustering process, each observation configuration is validated using a random forest classification algorithm

(Breiman, 2001). Furthermore, the two experiments are compared with an observation configuration that was operationally used within the regional CAMS reanalysis for 2016 (REF experiment). For this, the same data filtering as described above is applied. The REF experiment contains the same observation stations as the KSC and CD experiments. The experiments differ only in the assignment of the observation stations to the assimilation/validation set used in the data assimilation runs. Note that the CAMS REF configuration applied a bundling of $NO_2$ with $O_3$ and $PM_{10}$ with $PM_{2.5}$ observations (i.e., these species are

always assigned to the same data set) before selecting semi-randomly, where validation stations are placed near assimilation stations. Here, spatially isolated observations are used for the assimilation.

The impact of the station clustering on the representativity of the assimilation and validation data sets is evaluated using the 4D-var data assimilation system of EURAD-IM (Elbern et al., 2007). The data assimilation procedure optimizes the initial conditions as well as the emission factors for each grid box to obtain an improved analysis of the atmospheric state. The

225 evaluation is performed on data from four different months in 2016 (January, March, June, and September) to identify potential seasonal effects. The assimilation is performed on 30 consecutive days within each month with an assimilation window of 24 hours.

## 3.4 Definition of the representativity measure

The difference in the RMSE (Root-Mean-Square Error) of the analysed model state calculated for the assimilation and valida-

230 tion observations (hereafter AV-difference) is assumed as representativity measure with low values indicating a higher degree of representativity. A set of available observations (OBS) is split into an assimilation and validation set $(A + V)$, hereafter

observation configuration. To evaluate the split in terms of representativity, consider two distinct observation configurations with assimilation sets ($A_1$ and $A_2$) and validation sets ($V_1$ and $V_2$), with

$$\text{OBS} \rightarrow \begin{cases} A_1 + V_1, \\ A_2 + V_2. \end{cases} \tag{8}$$

Each assimilation set $A_i$ is used in the data assimilation system $M$ (e.g., the EURAD-IM as in this study), generating an analysis $P_i$

$$M : A_1 \mapsto P_1 \tag{9}$$
$$M : A_2 \mapsto P_2. \tag{10}$$

Note that $P_1 \neq P_2$ due to the different observation configurations. Amongst several possible options, the RMSE is chosen as 240 an error metric. Here, the RMSE of the analysis is calculated for each simulation day for the assimilation and validation sets for individual hourly data pairs.

$$E_i^A := \text{RMSE}(P_i, A_i) \tag{11}$$
$$E_i^V := \text{RMSE}(P_i, V_i) \qquad \text{for } i \in \{1, 2\}. \tag{12}$$

Then, the AV-difference is the absolute difference between $E_i^A$ and $E_i^V$. To compare the observation configurations, the 245 mean value of the AV-difference is calculated over the simulated days:

$$AV_i^{diff} := \overline{|E_i^A - E_i^V|}. \tag{13}$$

A perfect representativity is achieved if the AV-difference vanishes, i.e., the RMSE of the analysis compared to the assimilation data is equal to the RMSE compared to the validation data. A low AV-difference indicates that the modeled concentrations at the assimilation stations are representative for the concentrations in unobserved regions, where validation stations are located. 250 In turn, a high AV-difference shows an overfitting (or overconfidence) in observed areas compared to other regions. The model state is thus less representative, which limits the reliability of the analysis. The aim of this study is to evaluate the effect of different observation configurations on the representativity of the analysed atmospheric state. Although the analysis includes estimates of emission corrections of air pollutants, the evaluation of the accuracy of these corrections is not considered in this study.

An evaluation of the analysis' results in January shows that inaccuracies in the meteorological fields (especially in the wind fields) can lead to unrealistically strong emission corrections in certain locations. This is caused by a lateral displacement of the emission plume due to the erroneous meteorology, which results in a large RMSE at the surrounding observation stations. To account for this, stations with a RMSE larger than $4500\,\mu\text{g m}^{-3}$ as well as stations with emission correction factors larger than 15 in their vicinity are excluded from the evaluation of the representativity. These thresholds are chosen empirically

by investigating the model analysis and remove $\sim 2\,\%$ of observations. Both filtering criteria are rigorously applied for all analysed months. However, the former only applies in the January analysis. The emission correction factor based filtering affects all months for locations with a priori small emission strengths.

**Table 1.** Summary of the filtering procedures and the number of removed time series from the evaluation in percent.

| Method | Removed time series [%] |
|---|---|
| Outlier removal | 0.0 |
| Relative standard deviation | <0.1 |
| Constant intervals | 0.4 |
| Annually averaged diurnal cycle outlier | 0.3 |
| RMSE outlier (January and CO only) | 0.8 |
| Emission factor outlier | 2.2 |

## 4 Results

### 4.1 Analysis of the data clustering

The grouping of observation stations into clusters for the CD and KSC experiments is shown in Fig. 3. The locations of the observational sites for each pollutant species are shown in Fig. 1. Further, Fig. 4 shows the normalized mean and variance of the annually averaged diurnal cycle for the two clusters of the CD experiment for the observed species.

In the CD experiment, $86\,\%$ of the stations (1349 stations in total) are assigned to cluster 2. This cluster contains observations with generally lower mean and variance of the annually averaged diurnal cycle (Fig. 4) than observations in cluster 1 (227 stations), except for $O_3$. However, there is no spatial pattern in the two clusters since the geographical locations of the observation stations are not included as feature in the clustering process.

The clusters resulting from the KSC experiment are predominantly influenced by geographical location (Fig. 3). However, the boundaries between the clusters are not clearly defined, which would be seen in a k-means clustering that only takes the geographic location as features. This signifies the influence of the annually averaged diurnal cycle on the clustering results. Some agreement of the cluster structure with the Köppen-Geiger classification of climate zones in Europe (Peel et al., 2007) is found, showing that the clusters selected by the clustering algorithm are meaningful. Besides the geographical location, the clustering is influenced by the altitude of the observation station and the annually averaged diurnal cycle. This is illustrated in Fig. 5 for the altitude distribution for all clusters and in Fig. 6 for the annually averaged diurnal cycle of $NO_2$ averaged for all clusters used in the KSC experiment. Observation stations in clusters 4 and 7 show higher mean altitudes of $1024\,\mathrm{m}$ and $964\,\mathrm{m}$, respectively, while stations in the other clusters are generally located below $500\,\mathrm{m}$. Cluster 4 is mostly located in the central region of Turkey and cluster 7 is retrieved in the Alps and other mountainous regions mainly in Central Europe. Due to the

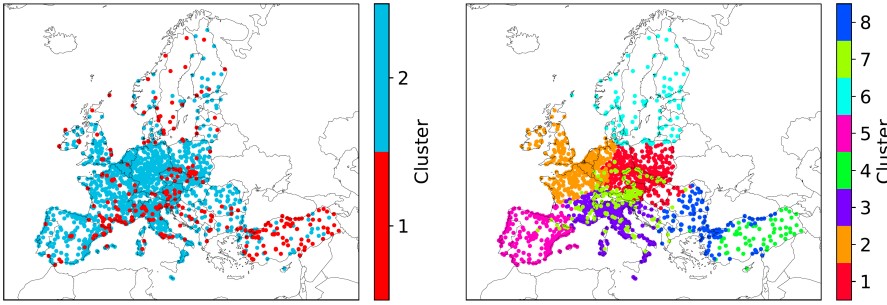

**Figure 3.** Distribution of the clusters for the CD (left) and KSC (right) experiment considering all observational sites in Europe regardless of the observed species. Different clusters are grouped by color.

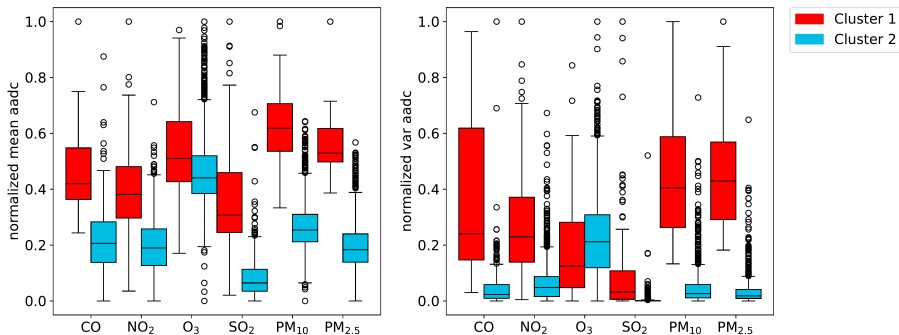

**Figure 4.** Comparison of the normalized mean (left) and variance (right) of the annually averaged diurnal cycle (aadc) per cluster generated in the CD experiment for each observed species. The boxes extend from the first to third quartile, the median is marked by the solid line, the whiskers extend to $1.5 \times$ the interquartile range. Outliers are marked as circles.

higher altitude of the observation stations in cluster 7, the annually averaged diurnal cycle shows the lowest mean ($10.9 \, \mu g \, m^{-3}$) and variance ($3.3 \, (\mu g \, m^{-3})^2$) for $NO_2$ concentrations among all clusters. However, the $NO_2$ mean diurnal cycle in this cluster is similar to that of cluster 6 (mean $12.3 \, \mu g \, m^{-3}$ and variance $4.5 \, (\mu g \, m^{-3})^2$), which mostly covers Scandinavia and therefore

includes mostly remote stations. The clusters 1, 2, 3, and 5 show a similar mean and variance of the annually averaged diurnal cycle for $NO_2$ concentrations, while clusters 4 and 8 show the highest values (mean: $37.2 \, \mu g \, m^{-3}$; $28.5 \, \mu g \, m^{-3}$; variance: $75.8 \, (\mu g \, m^{-3})^2$ $23.3 \, (\mu g \, m^{-3})^2$). This illustrates the ability of the clustering used in the KSC experiment to group stations according to the atmospheric chemical observations.

## 4.2 Evaluation of the representativity

The RMSEs for the assimilation and validation data sets for the REF, CD and KSC experiment are exemplary shown in Fig. 7 for 04 March 2016. Further, the AV-difference and the relative AV-difference is given for each experiment. The latter is calculated by dividing the AV-difference by the mean of the assimilation and validation RMSE. The REF experiment shows the

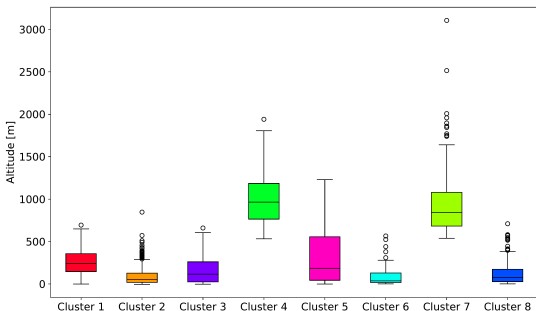

**Figure 5.** Distribution of the stations' altitude for all clusters obtained in the KSC experiment. The boxes extend from the first to third quartile, the median is marked by the solid line, the whiskers extend to $1.5 \times$ the interquartile range. Outliers are marked as circles.

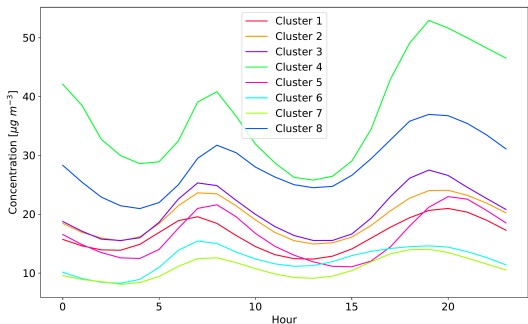

**Figure 6.** Annual mean diurnal cycles of the $NO_2$ concentration for all clusters generated in the KSC experiment. Each curve represents the annually averaged diurnal cycle for all stations within one cluster.

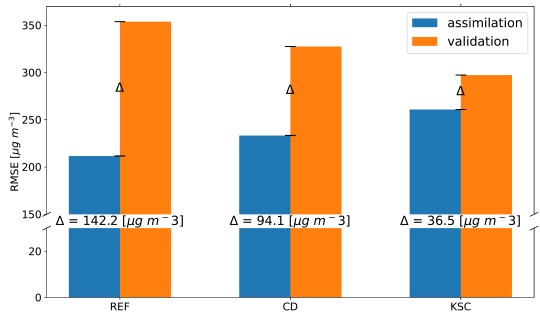

**Figure 7.** Comparison of the RMSE for CO of the assimilation and validation data sets for 04 March 2016, for the REF, CD and KSC experiment. The central number corresponds to the AV-difference for each experiment.

lowest RMSE for the assimilation data set ($211.7\,\mu g\,m^{-3}$) but the highest RMSE for the validation data set ($353.8\,\mu g\,m^{-3}$). This likely indicates an overfit of the analysis to the assimilation data or, in other words, a weak representation of unobserved regions. The observation configuration used in the KSC experiment yields the lowest AV-difference of $36.5\,\mu g\,m^{-3}$ for the CO concentration and thus provides the most representative model state. This corresponds to a relative AV-difference of 15 %. Furthermore, the figure illustrates that a representative model output is a trade-off between an increase in the RMSE for the assimilation data set and a decrease in the RMSE for the validation data set.

In order to asses the representativity for the whole assimilation period, the values of the mean AV-difference are averaged over all analysed days for all months. The results for CO and $NO_2$ are discussed in the following.

**Table 2.** AV-difference and RMSE for the assimilation and validation data sets for CO and $NO_2$ averaged over all simulated days in 2016 for the REF, CD, and KSC experiments.

| CO | $\overline{\text{RMSE}}_{\text{assim}}\;[\mu g\,m^{-3}]$ | $\overline{\text{RMSE}}_{\text{val}}\;[\mu g\,m^{-3}]$ | $\overline{\text{AV-difference}}$ / day $[\mu g\,m^{-3}]$ | relative $\overline{\text{AV-difference}}$ |
|---|---|---|---|---|
| REF | 240.3 | 324.4 | 87.0 | 31 % |
| CD | 272.8 | 290.6 | 49.5 | 18 % |
| KSC | 264.1 | 295.4 | 40.6 | 15 % |
| $NO_2$ | $\overline{\text{RMSE}}_{\text{assim}}\;[\mu g\,m^{-3}]$ | $\overline{\text{RMSE}}_{\text{val}}\;[\mu g\,m^{-3}]$ | $\overline{\text{AV-difference}}$ / day $[\mu g\,m^{-3}]$ | relative $\overline{\text{AV-difference}}$ |
| REF | 16.2 | 17.8 | 1.6 | 9 % |
| CD | 16.6 | 17.0 | 0.7 | 4 % |
| KSC | 16.7 | 17.3 | 0.8 | 5 % |

Table 2 compares the observation configurations from the different experiments in terms of the mean RMSE for the assimilation and validation data sets and the mean AV-difference considering all simulation days. The mean AV-difference is calculated using the absolute value of the differences on each day, thus,

$$\overline{\text{RMSE}}_{\text{assim}} - \overline{\text{RMSE}}_{\text{val}} \neq \overline{\text{AV-difference}}. \tag{14}$$

The evaluation of the mean AV-difference for CO shows a significant improvement in the representativity with a reduction from $87.0\,\mu g\,m^{-3}$ (relative AV-difference of 31 %) in the REF experiment to $49.5\,\mu g\,m^{-3}$ (18 %) and $40.6\,\mu g\,m^{-3}$ (15 %) in the CD and the KSC experiment configurations, respectively. The observation configuration in the KSC experiment yields the largest improvement with a decrease of the AV-difference of $\sim 53$ % which corresponds to a reduction in the relative AV-difference by 16 %. This indicates a reduction of the overfitting of the analysis.

The RMSE for $NO_2$ is one order of magnitude smaller than for CO with a mean assimilation RMSE of $16.2\,\mu g\,m^{-3}$ in the REF experiment. The same trend of an increasing mean RMSE for the assimilation data set and a decreasing mean RMSE for the validation data set can be seen, which leads to a reduction of the AV-difference for $NO_2$ of $\sim 56$ % (CD) and $\sim 50$ % (KSC). The decrease in the relative AV-difference is less substantial with a decrease of 5 % for the CD experiment and a decrease of 4 % for the KSC experiment. This is due to the generally lower RMSE for $NO_2$ than for CO.

The RMSE for $NO_2$, $O_3$, $SO_2$, $PM_{10}$, and $PM_{2.5}$ are an order of magnitude smaller than those for CO, and are therefore deemed to be of lesser significance for the evaluation of the configuration in the different experiments as the absolute difference is smaller. This is reflected in the relative AV-differences, which exhibit lesser changes compared to CO. Furthermore, the RMSE target reference values in the quality control of the CAMS regional services are 16 $\mu g\,m^{-3}$ for $O_3$, $PM_{10}$ and $PM_{2.5}$ and 22 $\mu g\,m^{-3}$ for $NO_2$ (Gauss et al., 2024), making AV-differences of $\sim 1\ \mu g\,m^{-3}$ negligible. The corresponding values of

the mean RMSEs and AV-differences for the other species are given in Table C1 in the Appendix.

    Note that the sum of $\overline{RMSE}_{assim} + \overline{RMSE}_{val}$ shows no significant difference between the observation configurations. This indicates that the quality of the analysis is not affected by the different experiments. Furthermore, while the evaluation shows fluctuations for each season, the general result holds true for each season individually, see Fig. A1 in the Appendix.

### 4.3 Validation of the experiments for the year 2017

To evaluate the generalizability of the findings in the experiments, the generated data split for the year 2016 is applied to data from March 2017. Here, the generated assimilation-validation data split that has been derived for the 2016 data is used to divide the 2017 data. Thus, the data clustering is not influenced by the data from 2017. Stations that are not included in both data sets are excluded to ensure comparability of the results. This leads to a reduction of the number of stations of approximately $9\,\%$ of observations in the assimilation data set.

**Table 3.** AV-difference and RMSE for the assimilation and validation data sets for CO and $NO_2$ averaged for March 2017 for the REF and the KSC experiment.

| CO | $\overline{RMSE}_{assim}\ [\mu g\,m^{-3}]$ | $\overline{RMSE}_{val}\ [\mu g\,m^{-3}]$ | $\overline{AV\text{-difference}}$ / day $[\mu g\,m^{-3}]$ | relative $\overline{AV\text{-difference}}$ |
|---|---|---|---|---|
| REF | 233.9 | 265.3 | 35.3 | 14 % |
| KSC | 240.2 | 250.4 | 32.1 | 13 % |
| $NO_2$ | $\overline{RMSE}_{assim}\ [\mu g\,m^{-3}]$ | $\overline{RMSE}_{val}\ [\mu g\,m^{-3}]$ | $\overline{AV\text{-difference}}$ / day $[\mu g\,m^{-3}]$ | relative $\overline{AV\text{-difference}}$ |
| REF | 16.6 | 18.8 | 2.3 | 13 % |
| KSC | 17.1 | 17.7 | 0.8 | 5 % |

Table 3 shows the mean RMSE for assimilation and validation stations and the mean AV-difference for CO and $NO_2$ concentrations in March 2017 for the REF and KSC experiments. To save computing time, the CD experiment has been excluded from this validation. For CO, the KSC experiment shows an improved overall representativity of the model analyses by $\sim 9\,\%$ for March 2017. This is likely due to the geographic information included in the KSC clustering algorithm, which is independent of the actual observations. However, this improvement is smaller than for the periods in 2016. Compared to

the average AV-difference for 2016 ($87.0\,\mu gm^{-3}$ per day), the mean AV-difference for the REF experiment is significantly lower in 2017 ($35.3\,\mu gm^{-3}$ per day), leaving less potential for improvement. This is reflected in the relative AV-difference, where an insignificant change of $1\,\%$ can be observed. The reduction of the mean AV-difference between the REF and the KSC experiments for $NO_2$ is more similar to the reduction obtained for the 2016 periods with $\sim 65\,\%$ reduction in March 2017

compared to $\sim 50\,\%$ in 2016. With a reduction in the relative AV-difference by $8\,\%$. An overview of the results for the other species is given in Table D1 of the Appendix.

The split of the available observation data derived for 2016 cannot be carried over directly to the year 2017. This is expected since clustering is inherently sensitive to the addition and removal of individual objects. Furthermore, the characteristics of observation stations are not static, as shown in Fig. B1 and Fig. B2 in the Appendix.

## 5   Summary and conclusions

In this study, two methods to choose representative assimilation and validation sets are developed and tested. They utilize k-means clustering algorithms and variations thereof and only require input from observations. Thus, they are independent of the assimilation model. The methods are tested in the CD and KSC experiment, where in the former, only the diurnal cycles of the observed concentrations are considered, and the latter experiment additionally accounts for the geographical locations and altitudes of the observational sites. They are compared to a REF experiment based on an observation configuration provided within the CAMS project.

The results demonstrate that the observation configurations derived in the experiments influence the analysis in data assimilation applications, which were conducted using the EURAD-IM. The quality of the analysis obtained using a configuration is quantified by its representativity. Here, the relative representativity of two datasets is determined by quantifying and subsequently comparing the difference in the RMSEs between the model analysis and the observations from the assimilation and validation data set. Investigating each month separately confirms these results, showing that the generated configurations outperform the split in the REF experiment. This is most notable for CO, where a limited number of observations is available. The clustering method used in the KSC experiment is capable of identifying clusters of observation stations that show different atmospheric chemical properties. Improvements in the representativity for CO ($NO_2$) are determined with a $\sim 53\,\%$ ($\sim 50\,\%$) reduction of the difference between the RMSE of the analysis compared to the assimilation and validation data set, which is regarded as the representativity measure. To account for the difference in the magnitude of the RMSE for the different species, the representativity measure is scaled with the average RMSE of the species. This yields a $16\,\%$ improvement for CO and only a $4\,\%$ improvement for $NO_2$. The less computationally demanding split in the CD experiment is also found to enhance the representativity, although to a lesser extend than in the KSC experiment. The clusters in the KSC experiment contain more information about the characteristics of their assigned observation stations (e.g., diurnal cycle and station altitude). Therefore, the KSC experiment is preferred for the selection of representative assimilation and validation data sets. However, the representativity for the other modelled species is not improved. This is due to the relative scarcity of the CO observations (see Fig. 1) and a resulting strong influence on the clustering. Furthermore, all species observations from a specific measurement site are categorized either into the assimilation or the validation set, called bundling. This is done to prevent advantages in the data assimilation due to the correlation between different species. A less restrictive bundling, as for example in the CAMS observational data set (REF experiment), could mitigate this disadvantage. In the REF experiment $NO_2$ is bundled with $O_3$ and $PM_{10}$ is bundled with $PM_{2.5}$.

The results have been obtained by considering observations for the entire year of 2016. This allows to generate one clustering result that can be applied to all analysis episodes within this year. It is assumed that a clustering using only data from the assimilation time period will further improve the representativity as the the stations' characteristics are considered more accurate. However, this will require more computational resources. Nonetheless, the results clearly show that a clustering on a broader period already provides benefits in terms of representativity compared to the observation configuration in the REF experiment.

The KSC experiment shows an improved representativity ($\sim 9\%$ compared to the REF experiment for CO) for an observation period with data from 2017 that has been split by copying the result from 2016 without subjecting the 2017 data to the clustering algorithm. The decrease in the degree of improvement found in the comparison to the 2016 period is attributed to the fact that the diurnal cycles of the observations in 2016 are not representative for the measurements made in 2017 (see Appendix B). Thus, the clustering and split are not optimal for the assimilation period compared to a data split that considers the observations from 2017.

A distinct advantage of the proposed method to generate observation configurations is that it guarantees the availability of validation data in all regions (clusters), which is crucial for species with sparse observations. It is worth noting that a k-means procedure that is solely based on longitude, latitude, and altitude of the stations can yield a clustering configuration similar to that of the cluster derived using the k-means soft constrained algorithm. However, in the case of localized variable mean concentrations of the observed species within one region, the k-means soft constrained algorithms is of advantage as it takes this difference into account. It is also important to consider the occurrence of extreme events, which may not be adequately captured by a purely geographical k-means approach.

Further investigations are envisioned to gain a deeper understanding of the influence of individual observation stations on the performance of the data assimilation algorithm (one aspect of observability) and the inclusion of other types of observations to optimize optimize the knowledge return from observation networks. Satellite observations are another common and valuable source of information for data assimilation systems. Extending the presented method to analyse the influence of these additional data on the representativity potentially provides insight into the optimal application of the data for assimilation.

*Code availability.* The routines for the k-means soft constraint clustering, including the data preparation can be found at https://doi.org/10.5281/zenodo.14711881 (last access: 07 August 2025). Instructions for running the code are included in the readme file (Hermanns, 2025).

# Appendix A: Seasonality of the AV-difference

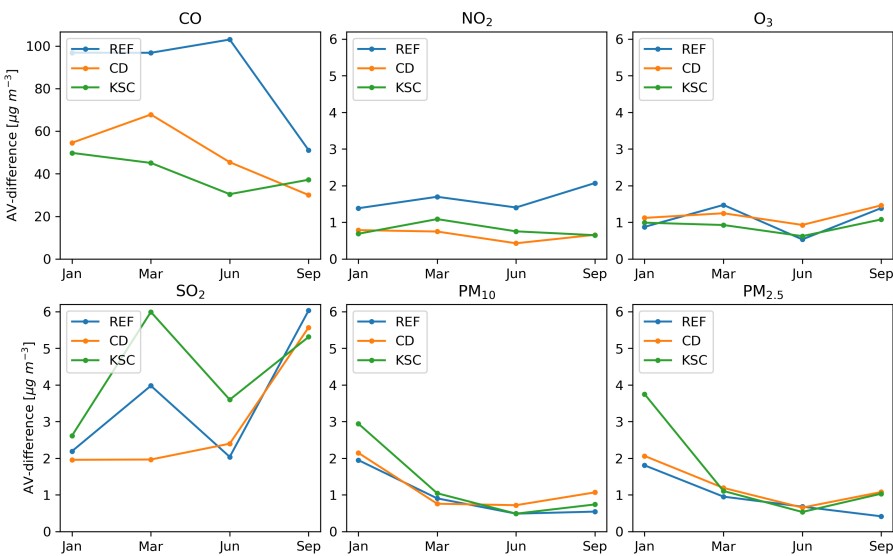

**Figure A1.** The evolution of the AV-difference for the REF, CD and KSC configurations for each species is shown over the four evaluated months of 2016.

 **Appendix B: Comparison of data from 2016 to 2017**

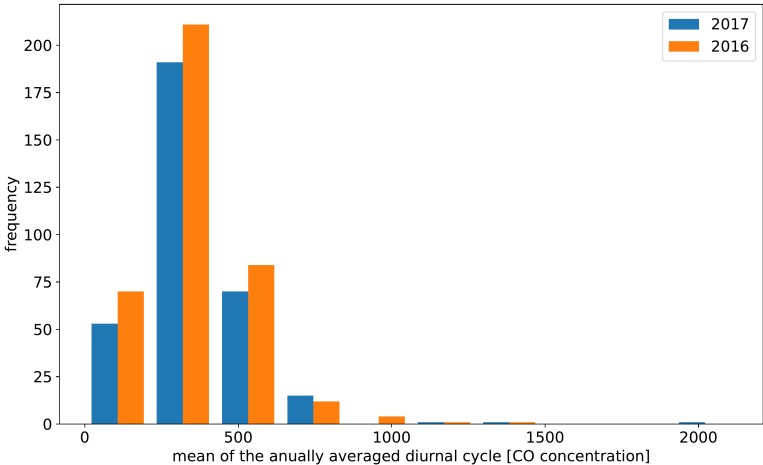

**Figure B1.** Histogram of the mean of the annually averaged diurnal cycles for CO observations in 2016 and 2017.

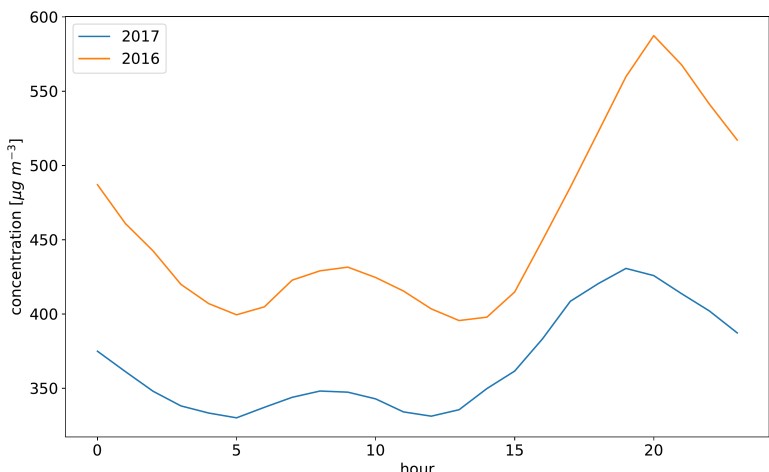

**Figure B2.** Annually averaged diurnal cycle for CO from the observation station "PL0575A" in 2016 and 2017.

# Appendix C: Mean AV-differences in 2016 for $SO_2$, $O_3$, $PM_{10}$, and $PM_{2.5}$

**Table C1.** AV-difference and RMSE for the assimilation and validation data sets for $SO_2$, $O_3$, $PM_{10}$ and $PM_{2.5}$ averaged over all simulated days in 2016 for the different observation configurations.

| $SO_2$ | $\overline{RMSE}_{assim}$ $[\mu g\ m^{-3}]$ | $\overline{RMSE}_{val}$ $[\mu g\ m^{-3}]$ | $\overline{AV\text{-difference}}$ / day $[\mu g\ m^{-3}]$ | relative $\overline{AV\text{-difference}}$ |
|---|---|---|---|---|
| REF | 10.4 | 10.4 | 3.6 | 35 % |
| CD | 10.7 | 8.7 | 3.0 | 31 % |
| KSC | 9.5 | 12.6 | 4.4 | 40 % |
| $O_3$ | $\overline{RMSE}_{assim}$ $[\mu g\ m^{-3}]$ | $\overline{RMSE}_{val}$ $[\mu g\ m^{-3}]$ | $\overline{AV\text{-difference}}$ / day $[\mu g\ m^{-3}]$ | relative $\overline{AV\text{-difference}}$ |
| REF | 22.5 | 23.5 | 1.1 | 5 % |
| CD | 22.2 | 23.4 | 1.2 | 5 % |
| KSC | 22.7 | 23.6 | 0.9 | 4 % |
| $PM_{10}$ | $\overline{RMSE}_{assim}$ $[\mu g\ m^{-3}]$ | $\overline{RMSE}_{val}$ $[\mu g\ m^{-3}]$ | $\overline{AV\text{-difference}}$ / day $[\mu g\ m^{-3}]$ | relative $\overline{AV\text{-difference}}$ |
| REF | 14.3 | 14.7 | 1.0 | 7 % |
| CD | 14.0 | 14.7 | 1.2 | 8 % |
| KSC | 14.6 | 14.2 | 1.3 | 9 % |
| $PM_{2.5}$ | $\overline{RMSE}_{assim}$ $[\mu g\ m^{-3}]$ | $\overline{RMSE}_{val}$ $[\mu g\ m^{-3}]$ | $\overline{AV\text{-difference}}$ / day $[\mu g\ m^{-3}]$ | relative $\overline{AV\text{-difference}}$ |
| REF | 11.0 | 11.3 | 1.0 | 9 % |
| CD | 10.5 | 11.3 | 1.2 | 11 % |
| KSC | 10.7 | 11.9 | 1.6 | 14 % |

# Appendix D: Mean AV-differences in 2017 for $SO_2$, $O_3$, $PM_{10}$, and $PM_{2.5}$

**Table D1.** AV-difference and RMSE for the assimilation and validation data sets for $SO_2$, $O_3$, $PM_{10}$ and $PM_{2.5}$ averaged over all simulated days in 2017 for the REF and KSC observation configurations.

| $SO_2$ | $\overline{RMSE}_{assim}$ $[\mu g\ m^{-3}]$ | $\overline{RMSE}_{val}$ $[\mu g\ m^{-3}]$ | $\overline{AV\text{-difference}}$ / day $[\mu g\ m^{-3}]$ | relative $\overline{AV\text{-difference}}$ |
|---|---|---|---|---|
| REF | 6.8 | 6.8 | 2.1 | 31 % |
| KSC | 6.0 | 8.1 | 3.0 | 43 % |
| $O_3$ | $\overline{RMSE}_{assim}$ $[\mu g\ m^{-3}]$ | $\overline{RMSE}_{val}$ $[\mu g\ m^{-3}]$ | $\overline{AV\text{-difference}}$ / day $[\mu g\ m^{-3}]$ | relative $\overline{AV\text{-difference}}$ |
| REF | 20.3 | 21.2 | 0.9 | 4 % |
| KSC | 20.3 | 21.3 | 1.1 | 5 % |
| $PM_{10}$ | $\overline{RMSE}_{assim}$ $[\mu g\ m^{-3}]$ | $\overline{RMSE}_{val}$ $[\mu g\ m^{-3}]$ | $\overline{AV\text{-difference}}$ / day $[\mu g\ m^{-3}]$ | relative $\overline{AV\text{-difference}}$ |
| REF | 13.2 | 14.0 | 1.0 | 7 % |
| KSC | 13.1 | 13.4 | 0.8 | 6 % |
| $PM_{2.5}$ | $\overline{RMSE}_{assim}$ $[\mu g\ m^{-3}]$ | $\overline{RMSE}_{val}$ $[\mu g\ m^{-3}]$ | $\overline{AV\text{-difference}}$ / day $[\mu g\ m^{-3}]$ | relative $\overline{AV\text{-difference}}$ |
| REF | 10.1 | 10.3 | 0.7 | 7 % |
| KSC | 9.9 | 10.7 | 1.0 | 10 % |

*Author contributions.* Contributed to conception and design: AH, PF, ACL. Contributed to acquisition of data: AH. Contributed to analysis and interpretation of data: AH, PF, ACL, JK, HF. Drafted and/or revised the article: AH, PF, ACL, JK, HF. Approved the submitted version for publication: AH, PF, ACL, JK, HF.

*Competing interests.* The authors declare to have no competing interests.

*Acknowledgements.* This work was partially performed as part of the Helmholtz School for Data Science in Life, Earth and Energy (HDS-LEE) and received funding from the Helmholtz Association of German Research Centres. The authors gratefully acknowledge the Earth System Modelling Project (ESM) for funding this work by providing computing time on the ESM partition of the supercomputer JUWELS (Jülich Supercomputing Centre, 2021) at the Jülich Supercomputing Centre (JSC).

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
