# Peer review of "Data clustering to optimise the representativity of observational data in air quality data assimilation: a case study with EURAD-IM (version 5.9.1 DA)"

_EGUsphere, 2025_

## Author Comment (AC1)

We would like to thank the reviewers for their thorough work and helpful feedback, which led to a significant improvement of the manuscript. In the following, all remarks by the reviewers are listed in black text and our corresponding replies are given in blue.

**Reviewer 1**

GENERAL COMMENT

The manuscript describes a method to distribute observation time series over clusters, and to use this within the context of data assimilation. The method is applied with the regional air quality model EURAD-IM, which assimilates time series of surface observations to guide the model. Specifically, the clustering is used to sub-divide the observation time series in an "assimilation" subset (~70%, incorporated in the assimilation) and a "validation" subset (~30%, not incorporated). The posterior comparison between analyzed model state and observations should give the same statistics over the "assimilation" and "validation" set, but as shown by the manuscript too, the assimilation usually performs better over the "assimilated" set. The proposed clustering method improves the equality between the statistics over the "assimilation" and "validation" set, and is therefore of interest for all data-assimilation applications.

The clustering method is well described, and easy to follow also for readers without a background in clustering. The application is illustrated for the European air quality network. Especially the maps in Figure 2 and bar plots in Figure 3 are useful here, as they illustrate the result of the clustering and how it was achieved. The 8-clusters obtained with the KSC method shows for example the soft borders between geographical regions, which could not be obtained by simply clustering countries. As described by the authors, the map obtained with KSC shares characteristics with climate zones, which gives trust that the obtained clustering is also related to geographic properties.

The improvement in assimilation/validation (AV) statistics is illustrated based on comparison with CO and NO2 observations (Table 1). In general, RMSE over the assimilation set increases, while the RMSE over the validation set decreases, thus decreasing what is called here the AV-difference. This is a very important result, and shows the usefulness of the method. However, as table C1 shows, the AV-difference increases for most other considered species (SO2, PM2.5, and PM10), and depending on the clustering method, also for O3. These species are rather important for air quality, and one could argue that these are even more important than CO. Therefore, the method seems not immediately applicable yet in for example the CAMS assimilations in which EURAD-IM is included. Could the authors include a discussion on how the clustering could be improved such that the AV-difference is decreased for all chemical species? Are different features of the timeseries needed, for example based on rural/urban locations? Or should for example CO simply be excluded? For the current manuscript it is not necessary to add and evaluate new clustering configurations, but it would be useful to see some guidance for future work.

Thank you for emphasizing the need to be more specific about the results of the clustering method on other species than CO/NO$_2$. In the presented clustering methodology, all species observations from a specific measurement station are categorized into either the assimilation or the validation

set, called bundling. This is done to avoid introducing artificial advantages in the validation due to the correlation between simulated species. Due to the scarcity of CO observations, they have a large influence on the clustering result. In future work, not applying the bundling or defining a less restrictive bundling could improve the AV-difference for the other chemical species to a similar extent. This is, for example, done in the assimilation/validation split within the CAMS project, where only parts of the species are categorized similarly ($NO_2$/$O_3$ and $PM_{10}$/$PM_{2.5}$).

We have added the following paragraph to the discussion section:

"However, the representativity for the other modelled species is not improved. This is due to the relative scarcity of the CO observations (see Fig. 1) and a resulting strong influence on the clustering. Furthermore, all species observations from a specific measurement site are categorized either into the assimilation or the validation set, called bundling. This is done to prevent advantages in the data assimilation due to the correlation between different species. A less restrictive bundling, as for example in the CAMS observational data set (REF experiment), could mitigate this disadvantage. In the REF experiment $NO_2$ is bundled with $O_3$ and $PM_{10}$ is bundled with $PM_{2.5}$."

SPECIFIC COMMENT

Line 184: Could the method used by CAMS to distribute observations in assimilation/validation set be summarized here? At lines 326-327 an essential difference is discussed, it might be useful to mention that earlier too.

We agree that a summary of the CAMS selection process is useful to understand the differences in the experiments. We have added this summary to the manuscript:
"The REF experiment contains the same observation stations as the KSC and CD experiments. The experiments differ only in the assignment of the observation stations to the assimilation/validation set used in the data assimilation runs. Note that the CAMS REF configuration applied a bundling of $NO_2$ with $O_3$ and $PM_{10}$ with $PM_{2.5}$ observations (i.e., these species are always assigned to the same data set) before selecting semi-randomly, where validation stations are placed near assimilation stations. Here, spatially isolated observations are used for the assimilation."

The data processing requires many steps, for example outlier removal (lines 144-155), but also removal of stations extreme emission corrections in their vicinity (lines 216-221). It would be useful to summarize all selection criteria in for example a table, including the number (fraction) of removed stations.

Thank you for the comment. We have included the following text passage and table to the main text:

"A summary of the filtering procedures and their effect on the amount of observational data is shown in Table 1.

Table 1. Summary of the filtering procedures and the number of removed time series from the evaluation in percent.

| Method | Removed time series [%] |
|---|---|
| Outlier Removal | 0.0 |
| Relative standard deviation | <0.1 |
| Constant intervals | 0.4 |
| Annually averaged diurnal cycle outlier | 0.3 |
| RMSE outlier (January and CO only) | 0.8 |
| Emission factor outlier | 2.2 |

"

The KSC clustering is applied using a location future, which gives a result that collects stations in geographic regions (adjacent countries and/or regions in countries). The map in the right panel of Figure 2 shows that within such cluster there are sometimes small regions with a different classification, for example the Pyrenees are part of cluster 7. Would it make sens to add features based on these "exceptions", for example the altitude of a station?

We thank the reviewer to emphasize that our description was not comprehensive enough, here. Indeed, the altitude is part of the features in the KSC clustering. For example, it clearly contributes to the definition of cluster 7. We have updated our description of the features in the manuscript to make this point clearer.

"In the second experiment, the k-means soft constrained algorithm is applied, named KSC in the following. Here, the features are the geographical coordinates and the altitude for each measurement station as well as the mean and variance of the annual average diurnal cycle."

SPELL AND GRAMMER

Lines 99 and 106: "k" should be "K" as in Figure 1?

Yes, thank you for the comment. All "k" in the text have been changed to "K".

Line 126: should be "... some objects, $F_m$, such that ..."

Done.

Line 225: should be reference to Fig. A1 ?

We have updated the reference. According to another comment by reviewer #2, we have moved the corresponding figure from the appendix to the main body of the text.

Line 240: "the Alps"

Done.

Line 253: remove comma's ?

Done.

Line 304: ".. month .."

Done.

---

## Author Comment (AC2)

We would like to thank the reviewers for their thorough work and helpful feedback, which led to a significant improvement of the manuscript. In the following, all remarks by the reviewers are listed in black text and our corresponding replies are given in blue.

**Reviewer 2**

**Paper summary**

Thank you for providing these remarks and comments. We are confident that our answer led to a compelling and well-improved manuscript.

This manuscript discusses a method for sub-sampling observational data in the context of air quality data assimilation, which requires to prepare the observational data into two datasets, respectively assimilation and validation. The authors propose to use clustering algorithms to improve the representativity of the observations during such a sub-sampling. Their methodology has two practical advantages: on the one hand, it is independent from the assimilation model, and on the other hand, it only requires observational data as inputs.

To evaluate the benefits of their clustering approach, the authors introduce an AV-difference (assimilation/validation) metric, which is the difference between the RMSE (Root-Mean-Square Error) of the model w.r.t. the assimilation dataset and the RMSE of the model w.r.t. the validation datasets. As such, a AV-difference of zero is synonymous of perfect representativity, while a high AV-difference suggests overfitting by the model.

Using an operational CAMS assimilation/validation configuration for year 2016 as a reference, the authors apply their approach on observations over Europe for four months of year 2016 (January, March, June and September, picked for their seasonal representativity), and demonstrate a significant decrease in the AV-difference for several pollutant species, and most notably carbon monoxide (53%), nitrogen dioxide (50%) and ozone (18%). The improvement is particularily interesting in the case of carbon monoxide, due to the scarcity of the observations compared to other pollutant species (such as ozone).

**General comments**

The core content of the manuscript is interesting and provides some convincing results, and the authors did a nice job in presenting the K-means clustering algorithm and its soft constraint variant, and how they adapted their problem to both. This said, this manuscript could be improved in terms of presentation and could elaborate on a few points to ensure the final paper is compelling to all.

**Possible presentation improvements**

I found that the AV-difference metric was very important to understand the paper and its contributions, yet it's defined quite late in the manuscript (L193). The Introduction does make a review of the state-of-the-art in this regard, but only states that the present study will improve representativity through clustering without hinting at how it will measure it. A few sentences (if not a single one) in the Introduction to give the big picture may be enough.

We agree that highlighting the big picture of our analysis in the introduction will lead to an enhanced understanding of our results. We have added a description of the AV-difference as the representativity measure used in the paper to the introduction. This addition reads:

"Here, the relative representativity of two datasets is determined by quantifying and subsequently comparing the difference in the RMSEs between the model analysis and the observations from the assimilation and validation data set. This measure is hereafter called AV-difference and is described in detail in Sec. 3.4."

The manuscript could also benefit from a few more figures to support its content. A possible addition could be a flow-chart in the Introduction, summarizing the proposed methodology (e.g., observations going through the clustering to be split into the two datasets, fed to a data assimilation model like EURAD-IM). Such a flow-chart would not only summarize the overall methodology to the reader in a single figure, but could also be used to picture its advantages in terms of input/output.

Thank you very much for this valuable input.  We have added a flowchart (now Fig. 2) to give an overview over the proposed method. It details the process from the input data to the clustering up to the evaluation of the representativity that we employed in this manuscript. Further, we have added Fig. A1 to showcase the seasonality of the representativity enhancement to the manuscript as requested by reviewer #3.

The new Fig. 2:

[Figure]

Figure 2. Flowchart of the proposed methodology. In the presented case, 'Observation Data' is the observation data used in the CAMS-project. The elements in the box titled 'included in code' are included in the provided software code (Hermanns, 2025), with the KSC algorithm as the 'Clustering'. The box 'split' is the extraction of the assimilation and validation set from a clustering result. 'DA' is short for data assimilation. The 'AV-difference' is the representativity measure and detailed in Sec. 3.4.

The new Fig. A1:

[Figure]

Figure A1. The evolution of the AV-difference for the REF, CD and KSC configurations for each species is shown over the four evaluated months of 2016.

I would also recommend moving Figure A1 from the Appendix back to the main body. Indeed, Figure A1 gives a very clear picture of how scarce CO observations are with respects to other pollutant species. Including it into the main body and making a few more references to it would strenghten the conclusion that the proposed clustering methodology significantly improves representativity of CO in the framework of air quality data assimilation.

Thank for the recommendation. We agree that the figure is better placed in the main body to strengthen the argumentation. It is moved to chapter 3. We have added this additional statement in the main text of chapter 3:
"The density of CO observations compared to the other measured species is substantially lower, even in highly populated areas. "
And this statement to the Conclusion:
"This is due to the relative scarcity of the CO observations (see Fig. 1) and a resulting strong influence on the clustering."

Finally, on a side note, I would recommend using a gridded layout for most of the figures, especially line plots.

Thank you for the suggestion. We have generated the plots using a gridded layout. However, this

adds a lot of lines to the figures, which we find distracting from the main results. Therefore, we decided to leave the figures as they were.

**Questions regarding the content**

1) Are there particular reasons for only simulating four months of 2016 ? I get the seasonality argument regarding the choice of the months, but why not simulating the entire year ?

We agree that it would have been interesting to simulate and evaluate the proposed method on the entire year. However, the employed four-dimensional variational data assimilation methodology requires an iterative optimization of the parameters to be optimized (here, initial values and emission factors). Therefore, simulating an entire year is computationally very demanding. The simulations conducted for this manuscript are based on the simulations by Lange et al., 2023, where we identified issues with the representativity of the utilized observational split provided within the CAMS project. The simulations conducted here to evaluate the model's performance given the split of our proposed method aim to illustrate its potential while keeping the computational burden feasible.

2) More broadly, it would be interesting to develop the seasonality of the results. At L276, there is this mention:

>Furthermore, while the evaluation shows fluctuations for each season, the general result holds true for each season individually.

but this is not enough to convince the reader about the seasonal trends of the results, especially considering point 1) (i.e., no full seasons, only sample months) and considering there is no figure or table detailing seasonal results. If these trends are indeed not significant, maybe a single table or figure would be enough to demonstrate that.

Thank you for the suggestion. We have added a figure detailing the seasonal evolution (as given by the four simulated months) in the appendix, Fig. A1. It shows the evolution of the AV-difference for each species for each season. The new Fig. A1 is shown above.

3) What about the slightly worse results for KSC in Tables C1 and D1 (Appendices C and D) ? Should we worry about them or are they small enough to be ignored ? While there is, indeed, an order of magnitude of difference between these results and those for carbon monoxide, additional details could show decisively whether or not the slightly higher AV differences are problematic, and at the very least, why the current manuscript does not elaborate further on them.

* For instance, the slighter higher AV-difference for ozone in D1 is probably not much of an issue given the thresholds for air quality. E.g., below 80 µg per cubic meter of ozone is considered to be good per CAMS, so 1.1 µg per cubic meter of AV-difference remains negligible. However, the reader does not necessarily know about such orders of magnitude depending on the species.

Thank you for this suggestion. We have included a reference to the RMSE target reference values in the CAMS quality control. "Furthermore, the RMSE target reference values in the quality control of the CAMS regional services are 16 µg m$^{-3}$ for O$_3$, PM$_{10}$ and PM$_{2.5}$ and 22 µg m$^{-3}$ for NO$_2$ (Gauss et al., 2024), making AV-differences of ~ 1 µg m$^{-3}$ negligible.". We also reworked chapter 4 to include the relative AV-difference. This is calculated by dividing the mean AV-difference by the mean of the assimilation and validation RMSE. The relative AV-difference better highlights the impact of the improvement depending on the magnitude of the RMSE.

* As far as I'm concerned, I would be also interested in learning if the given AV-differences are constant throughout each year (i.e. 2016 or 2017), or if they depend on the season, if not the day ? A plot of the AV-difference throughout the year for each species may be enough to address this concern.

Thank you for making this important statement. Indeed, we have analyzed the temporal evolution of the AV-difference and have decided to leave this discussion out of the manuscript to be more concise in our results. However, the reviewer comments suggest that the additional evaluation of the temporal evolution of the AV-difference increases the clarity of our results. Therefore, we have added a figure detailing the AV-differences during the seasons in the appendix (now Fig. A1). The AV-differences behave similarly for all months, although some variability on the daily AV-difference exist.

**Specific comments**

>L91: An overview of the geographic distribution of the available observation stations for each species is shown in the appendix in Fig. B2.

The reference seems to be wrong; the geographic distribution is shown in Fig. A1. Note that this overlaps with a previous comment on moving such figure back to the main text.

Done. We have put the Figure in the main text as requested by your previous comment.

>L128: "Is is termed to be violated [...]"

Done.

This looks like a typo. Shouldn't it be "It is termed to be violated..." ? Anyway, the full sentence is a bit unclear. What does "violated" mean precisely in this context ? Does it mean the sigma term only makes sense when the objects are assigned to distinct clusters ? Please clarify.

The term "violated" originates from Wagstaff, 2004. We agree that it is confusing in the presented context and does not serve the understanding. The term "violated" has been removed from the text and the descriptions altered accordingly.

---

## Author Comment (AC3)

We would like to thank the reviewers for their thorough work and helpful feedback, which led to a significant improvement of the manuscript. In the following, all remarks by the reviewers are listed in black text and our corresponding replies are given in blue.

**Reviewer 3**

Data clustering has the potential to improve the representativity of data assimilation results. This is shown by Hermanns and co-authors in their paper. This is an interesting idea and would be something to incorporate in for instance the European CAMS ensemble analysis and reanalyses. However, the paper also raised multiple questions and I am not yet convinced that the potential of the method has been fully exploited. To my opinion a major revision is needed in response to my major and detailed comments.

We thank the reviewer for his/her careful revision of our manuscript. We believe that incorporating the reviewers comments and suggestions led to an improved manuscript.

Major comments:

The results for PM2.5/10 (and ozone) should also be shown and should be discussed in more detail. Why is the result so different? Can this be understood? In the abstract, last sentence, improvements are reported for NO2, O3, CO: Please report the PM results as well.

We have reworked chapter 4 to include the relative AV-difference. This is calculated by dividing the mean AV-difference by the mean of the assimilation and validation RMSE. The relative AV-difference better highlights the impact of the improvement depending on the magnitude of the RMSE and explains the difference in the results. The abstract has been changed to report the relative AV-difference, including for $SO_2$, $PM_{10}$ and $PM_{2.5}$.
We have changed the last sentence of the Abstract as follows: "[...] the largest improvement in the relative representativity measure is evaluated for CO with 16 %, for $NO_2$ with 4 %, and for $O_3$ with 1 %. A reduction in the relative representativity measure is observed for $SO_2$ with -5 %, for $PM_{10}$ with -2 % and for PM2.5 with -5%, although these differences do not lead to significant deviations in absolute values given the overall error and the improvement for CO outweighing the changes in the other species."

The motivation for - and introduction to - the clustering approach can be improved. In particular I was wondering why the diurnal cycle is used as property to distinguish stations for all species? The diurnal cycle of ozone is large during summertime pollution photochemical smog events when it builds up during the day. For other species(like NO2, PM) the diurnal cycle may have a very different interpretation, e.g. rush hour emission peaks or development of the PBL. The effectiveness of this choice may be quite different in summer and winter. Apart from the diurnal cycle, are there other properties that may be used for the clustering? Please add a discussion to answer these questions, and also add the seasonal results.

We have added the corresponding Fig. A1 to the appendix and referenced it in:
"Furthermore, while the evaluation shows fluctuations for each season, the general result holds

true for each season individually, see Fig. A1 in the Appendix." The choice of parameters regarding the averaged diurnal cycle of air pollutants (mean and standard deviation) was made to find a compromise between the number of parameters used and the accuracy of the used parameters. As was shown by Beyer et al., 1999, the distance calculated within the KSC method approaches its maximum value as the dimension of the problem increases. Therefore, we wanted to limit the number of parameters to a minimum. Further, the parameters of our choice are closely related to the parameters chosen by, e.g., Gaubert et al. 2014, and Joly and Peuch, 2012, which made us confident that the mean and standard deviation of the averaged diurnal cycle leads to good results, as was confirmed by our discussion of the clustering. We have added the following information to the manuscript: "While other features are also suitable for clustering as was shown e.g., in Joly and Peuch (2012), restricting the number of features is important to apply meaningful clustering. In high dimensions, the "nearest neighbour" problem cannot be solved in all cases (Beyer et al., 1999)."

The new Fig. A1:

[Figure]

Figure A1. The evolution of the AV-difference for the REF, CD and KSC configurations for each species is shown over the four evaluated months of 2016.

Is the comparison with CAMS a fair comparison? Are the same stations used in both cases? The CAMS REF experiment is not well described in the paper and I have the impression that there are much less stations used by CAMS. Showing the distributions of assimilation and validation stations in all experiments could be a useful extra plot.

Thank you for pointing out that our description of the experimental setup was not sufficient. All presented experiments use the same observation stations as the CAMS REF experiment. The

configurations only differ in the assignment of individual observations to the assimilation or validation data set; thus, the same number of stations is used in all experiments, including CAMS REF. We have referenced the EEA data base for the general user who does not have access to the pre-selected data set from the CAMS REF experiment. We have updated the description of the observational data accordingly (see below). Anyway, we have decided to not include a plot for the individual observation splits as the large number of stations prohibit a detailed assessment of the assignment for single stations. The paragraph now reads: "The observational data used in this study are ground-based observations used within CAMS regional analysis. The data is pre-filter to the classes 1-7 according to Joly and Peuch 2012 from data accessible at the European Environmental Agency (EEA) via *https://eeadmz1-downloads-webapp.azurewebsites.net/* (last access: 09 June 2025). They consist of measurements of hourly concentrations of carbon monoxide (CO; 382 stations), nitrogen dioxide (NO2; 1743 stations), ozone (O3; 1712 stations), sulfur dioxide (SO2; 1005 stations), PM10 (1029 stations), and PM2.5 (509 stations) from 40 EEA member and cooperating countries."

The European stations also come with a site classification (rural/urban, background/traffic/industry). This by itself can be seen as a clustering. In CAMS the Joly-Peuch site classification is used (https://doi.org/10.1016/j.atmosenv.2011.11.025). Again, the categories 1-10 of Joly-Peuch are also a form of clustering. Please add this reference and discuss the relation to the present study.

Thank you for pointing out that we missed to put the proper citation to the manuscript. We have included a discussion of the paper and described its relevance for CAMS. However, we also like to highlight that our approach does not aim to classify stations directly but to identify and derive parameters that can be used to find a representative split to assimilation and validation sites. The new paragraph reads: "The study by Joly and Peuch (2012) characterizes pollutant time series by 8 parameters derived from their daily-, monthly-, seasonally- and annually averaged diurnal cycles. With this, they were able to derive 10 classes of air pollution monitoring sites using linear discriminant analysis, which allow for the classification of new sites without the need to recalculate the classification. This classification is used by the Copernicus Atmosphere Monitoring Service (CAMS) to filter observational data in order to improve their regional products (Peuch et al., 2022)"

The EURAD assimilation, if I understand well, has been run at a 15 km resolution. Sites near busy roads or near industries will not be well represented at this resolution. A pre-filtering would be good, as is done in CAMS. On the contrary, from the paper I get the impression that all EEA sites are used by the authors. Please explain and motivate this choice.

Thank you for this comment. As stated above, the same preselected stations as in CAMS are used. This excludes sites near busy roads or industries. We clarified this in the text: "The REF experiment contains the same observation stations as the KSC and CD experiments. The experiments differ only in the assignment of the observation stations to the assimilation/validation set used in the data assimilation runs. Note that the CAMS REF configuration applied a bundling of $NO_2$ with $O_3$ and $PM_{10}$

with PM$_{2.5}$ observations (i.e., these species are always assigned to the same data set) before selecting semi-randomly, where validation stations are placed near assimilation stations. Here, spatially isolated observations are used for the assimilation."

The filtering that needs to be used to account for unrealistic results in EURAD (line 216-221) gives an uneasy feeling. Please add evidence that the EURAD system is working well overall, with reasonable increments and good reductions of the rmse differences with the stations in the analysis. How has EURAD been tested?

The filtering for unrealistic results is only relevant for the month of January as the other simulated months do not show this behaviour. We have investigated the critical areas and found errors in the wind field used to disperse the emitted pollutants. Due to these errors, the emission plume was slightly offset to the observation location. Thus, the modelled plume only hit the observation location in its edges. The assimilation system (as any other presently used assimilation system) is not aware of plume displacements and correctly tried to increase the emissions. However, this was an exceptional situation. We chose to implement a rigorous filter to exclude this situation. Further, the EURAD-IM is continuously evaluated within the CAMS Regional Air Quality Production System (https://doi.org/10.5194/egusphere-2024-3744). Further, the results of the four-dimensional variational data assimilation system have been used in peer reviewed publications, e.g., Franke et al., 2023, Erraji et al., 2024.

I did not find the 2017 results fully satisfactory. One would hope that the clusters are quite robust and do not change much from year to year.

The classification based the clustering algorithm is inherently sensitive to the addition or removal of elements. Furthermore, the characteristics of observation stations used in the evaluation are not static. We consider it an advantage to not define a fixed classification for the stations but rather a modular one where the output is dependent on the available data. This can ensure to derive a representative assimilation set from the available observations. We have addressed your concerns in the text: "The split of the available observation data derived for 2016 cannot be carried over directly to the year 2017. This is expected since clustering is inherently sensitive to the addition and removal of individual objects. Furthermore, the characteristics of observation stations are not static, as shown in Fig. B1 and Fig. B2 in the Appendix."

Introduction:

- Introduction: Please add more references on air quality data assimilation activities. For instance the CAMS ensemble activity is relevant for this paper. Here also a split in observation and validation sites is applied (l 27).

We have added further references.

"The Copernicus Atmosphere Monitoring Service (CAMS, https://atmosphere.copernicus.eu/regional-air-quality-production-systems (last access: 07 August 2025)) employs an ensemble of air quality models using data assimilation to provide daily air quality forecasts and reanalysis over Europe. An overview of the ensemble and the models used within can be found in Marécal et al. (2015)"

- Introduction (l 31). Classification of surface sites is a topic related to the current paper. The paper of Joly-Peuch is a basis for the CAMS work and is relevant to discuss. Also the methodology of EEA should be mentioned (with reference).

We have added the reference to the Joly-Peuch paper including the relevance for the CAMS work (see also our comment above). Further, we have clarified the use of EEA data in the text, see also our answer to your comment above.

- l 58: Please provide a motivation why the diurnal cycle is used.

In Lyapina et al. (2016) the monthly averaged diurnal cycles were chosen as clustering parameters since they yield the most stable clustering results. We have chosen not to elaborate on their decision process since it would not benefit our paper.

- l 80: (IFS) please mention the European Centre of Medium-Range Weather Forecasts

Done

l 87: There is no pre-selection made for the stations? Would it be better to remove roadside traffic stations?

As mentioned above, we used the same data as in the CAMS assimilation system, thus the filtering according to the Joly-Peuch classification has been applied to the EEA data. We have added clarifications in the main text, see above.

- l 92: "An overview of the geographic distribution of the available observation stations for each species is shown in the appendix in Fig. B2" Do you mean A1?

We changed the reference in the text as we also moved the Figure to the main text as suggested by reviewer 2.

- Figure 1 is not a figure. A table could be an option, or a listing in the text would be possible as well.

We have moved the content of figure 1 into the text.

- l 104: Normalised: how? Also units should be removed for this formula to make sense (e.g. concentration).

We have added an explanation about the normalization of the feature vectors. The sentence now reads: "The elements of the feature vectors are normalized to a dimensionless value between [0,1] in order to ensure that each feature affects the distance value with the same weight."

- l 152: What is the standard deviation of the mean and variance of the diurnal cycle? A formula would be good to be precise on what is computed.

We have given a more detailed explanation in the text. We feel that a formula would hinder readability due to the complexity of the necessary indexing. The description now reads: "The annually averaged diurnal cycle is calculated from the filtered observational data by computing the average for each hour of the day for the entire year. From this annually averaged diurnal cycle, the mean and the variance are calculated and form one set of the features used in the clustering algorithms."

- l 184: "REF experiment" Please provide more details what this is and how it compares. Does CAMS include a similar number of stations? CAMS uses Joly-Peuch classification to pre-select the stations that are compared to the models.

In accordance with a previous remark, we have added more information about the REF experiment in the main text (see our comment above).

- l 193: The AV-difference: is this normalised? Does this have a unit (e.g. ug/m3)?

In line 193 the AV-difference has no unit, as it is derived in a general context and the unit depends on the evaluated quantities (RMSE in our case). Later, where the AV-difference is evaluated, the unit is given. It is not normalised.

- l 194: "split into two different observation configurations" This was unclear to me. Why is this done, and how are these two constructed?

Thank you for pointing this out. We added a more detailed explanation in the text: "A set of available observations (OBS) is split into an assimilation and validation set (A + V), hereafter observation configuration. To evaluate the split in terms of representativity, consider two distinct observation configurations with assimilation sets ($A_1$ and $A_2$) and validation sets ($V_1$ and $V_2$), [...]"

- l 203: Is the RMSE computed for individual model/measurement pairs (hourly observations)?

The RMSE is computed for each hourly model/measurement pair and for each day separately. We have added some clarifying remarks: "Amongst several possible options, the RMSE is chosen as an error metric. Here, the RMSE of the analysis is calculated for each simulation day for the assimilation and validation sets for individual hourly data pairs."

- Fig. B2: please add diurnal cycles for all species.

Thank you for the suggestion. Our point is that the annually averaged diurnal cycle of a single species at a single station may be different from year to year. Therefore, we have chosen the CO time series to illustrate this. We like to emphasize that this behaviour is possible but not necessarily true for all species/stations. Therefore, we would like to keep the figure as is. Adding more species, which may not change strongly among the years, will hamper readability and distracts from the point we want to make with this illustration.

---

## Author Response (AR2)

We would like to thank the reviewers for their work and helpful comments. They have helped to significantly improve the manuscript. In the following, the technical remarks are listed in black text and our corresponding replies are given in blue.

**Reviewer 3**

The authors have addressed my comments in a satisfactory way, and the discussion has been extended to other quantities including PM. The abstract is now more balanced. The introduction of the relative AV-difference is a good addition. The approach is now better explained, and properly refers to the CAMS activity and the Joly-Peuch paper.

Please add the latest publication for the CAMS ensemble as addition to the Marecal reference (even though it is a preprint at this moment): Colette et al., https://doi.org/10.5194/egusphere-2024-3744. Marecal et al is 10 years old and no longer up-to-date.

Thank you for again revising our manuscript and pointing out the more recent publication. We have added it to the text, as suggested.
The paragraph now reads: "An overview of the ensemble and the models used within can be found in Marecal et al. (2015) and Colette et al. (2025). "